# To Learn Effective Features: Understanding the Task-Specific Adaptation of MAML

## Abstract

Meta learning, an effective way for learning unseen tasks with few samples, is an important research area in machine learning. Model Agnostic Meta-Learning (MAML) (Finn et al. (2017)) is one of the most well-known gradient-based meta learning algorithms, that learns the meta-initialization through the inner and outer optimization loop. The inner loop is to perform fast adaptation in several gradient update steps with the support datapoints, while the outer loop to generalize the updated model to the query datapoints. Recently, it has been argued that instead of rapid learning and adaptation, the learned meta-initialization through MAML has already absorbed the high-quality features prior, where the task-specific head at training facilitates the feature learning. In this work, we investigate the impact of the task-specific adaptation of MAML and discuss the general formula for other gradient-based and metric-based meta-learning approaches. From our analysis, we further devise the Random Decision Planes (RDP) algorithm to find a suitable linear classifier without any gradient descent step and the Meta Contrastive Learning (MCL) algorithm to exploit the inter-samples relationship instead of the expensive inner-loop adaptation. We conduct sufficient experiments on various datasets to explore our proposed algorithms.

## 1 Introduction

Few-shot learning, aiming to learn from few labelled examples, is a great challenge for modern machine learning systems. Meta learning, an effective way for tracking this challenge, enables the model to learn general knowledge across a distribution of tasks. Various ideas of meta learning have been proposed to address the few-shot problems. Gradient-based meta learning (Finn et al. (2017); Nichol et al. (2018)) learns the meta-parameters that can be quickly adapted to new tasks by few gradient descent steps. Metric-based meta learning (Koch et al. (2015); Vinyals et al. (2016); Snell et al. (2017)) proposes to learn a metric space by comparing different datapoints. Memory-based meta learning (Santoro et al. (2016)) can rapidly assimilate new data and leverage the stored information to make predictions.

Model Agnostic Meta-Learning (MAML) (Finn et al. (2017)) is one of the most well-known gradient-based meta learning algorithms, that learns the meta-initialization parameters through the inner optimization loop and the outer optimization loop. For a given task, the inner loop is to perform fast adaptation in several gradient descent steps with the support datapoints, while the outer loop to generalize the updated model to the query datapoints. With the learned meta-initialization, the model can be quickly adapted to the unseen tasks with few labelled samples. Following the MAML algorithm, many significant variants (Finn et al. (2018); Rusu et al. (2018); Oreshkin et al. (2018); Bertinetto et al. (2018); Lee et al. (2019b)) are studied under the few-shot setting.

To understand how the MAML works, Raghu et al. (2019) conduct a series of experiments and claim that rather than rapid learning and adaptation, the learned meta-initialization has already absorbed the high-quality features prior, thus the representations after fine-tuning are almost the same for the coming unseen tasks. Also, the task specific head of MAML at training facilitates the learning of better features. In this paper, we further design more representative experiments and present a formal argument to explain the importance of the task specific adaptation. Actually, the multi-step task-specific adaptation, making the body and head have similar classification capabilities, can provide better gradient descent direction for the features learning of body. We also notice that for both the

gradient-based methods (e.g. MAML (Finn et al. (2017)), MetaOptNet (Lee et al. (2019b))) and metric-based methods (e.g. Prototypical Networks (Snell et al. (2017))) that attempt to learn a task-specific head using the support datapoints, the adaptation is a common mode for features learning of body but varied in different methods.

Based on our analysis, we first propose a new training paradigm to find a decision plane (linear classifier) for guidance with no gradient descent step during the inner loop and get more supporting conclusions. Moreover, we devise another training paradigm that removes the inner loop and trains the model with only the query datapoints. Specifically, inspired by contrastive representation learning (Oord et al. (2018); Chen et al. (2020); He et al. (2020)), we exploit the inter-samples relationship of query set to find a guidance for the body across different tasks. This meta contrastive learning algorithm even achieves competitive results comparable to some state-of-the-art methods. In total, our contributions can be listed as follows:

1. We present sufficient experiments and formal argument to explore the impact of the task-specific adaptation for body features learning and discuss the general formula for other gradient-based and metric-based meta-learning approaches.

2. We devise a training algorithm to obtain a decision plane with no gradient descent step during the inner loop, named as Random Decision Planes (RDP), and get more supporting conclusions.

3. Unlike prior gradient-based methods, we propose the Meta Contrastive Learning (MCL) algorithm to exploit the inter-samples relations instead of training a task-specific head during the inner loop. Even without the task-specific adaptation for guidance, our algorithm still achieve better results with even less computation costs.

4. We empirically shows the effectiveness of the proposed algorithm with different backbones on four benchmark datasets: miniImageNet (Vinyals et al. (2016)), tieredImageNet (Ren et al. (2018)), CIFAR-FS (Bertinetto et al. (2018)) and FC100 (Oreshkin et al. (2018)).

## 2 RELATED WORKS

MAML (Finn et al. (2017)) is a highly influential gradient-based meta learning algorithm for few-shot learning. The amazing experiment results on several public few-shot datasets have proved its effectiveness. Following the core idea of MAML, there are numerous works to handle the data insufficiency problem in few-shot learning. Some works (Oreshkin et al. (2018); Vuorio et al. (2019)) introduce the task-dependent representations via conditioning the feature extractor on the specific task to improve the performance. Sun et al. (2019) also employ the meta-learned scaling and shifting parameters for transferring from another large-scale dataset. Others (Grant et al. (2018); Finn et al. (2018); Lee et al. (2019a)) study this problem from the perspective of Bayesian approach. Unlike prior methods, we provide two training paradigms, one with no gradient descent step during the inner loop and another removing the inner loop and exploiting the inter-sample relations for training.

Recent works also explore the key factors that makes the meta-learned model perform better than others at few-shot tasks. Chen et al. (2019) discovers that a deeper backbone has a large effect on the success of meta learning algorithm, while Goldblum et al. (2020) finds that the meta learning tends to cluster object classes more tightly in feature space for those methods that fix the backbone during the inner loop (Bertinetto et al. (2018); Rusu et al. (2018)). A very recent work (Raghu et al. (2019)) argues that the meta-trained model can be applied to new task due to the high-quality features prior learned by the meta-initialized parameters rather than rapid learning. In this paper, we further study the impact of the task-specific adaptation for feature learning. Based on the analysis, we devise two algorithms, Random Decision Planes (RDP) and Meta Contrastive Learning (MCL) requiring less computation cost but still with competitive performance.

## 3 MODEL-AGNOSTIC META LEARNING (MAML)

The MAML aims to learn the meta-initialized parameters $\theta$ for the coming unseen tasks through the inner optimization loop and the outer optimization loop. Under the $N$-way-$K$-shot setting, for a task $T_b$ sampled from the task distribution $P(T)$, we have a support set of $N \times K$ examples $T_b^s$ and

Table 1: The evaluation results of 5-way-K-shot learning for methods with different training regimes on the MiniImageNet and TieredImageNet datasets.

| Method | MiniImageNet-5-way | TieredImageNet-5-way |
|---|---|---|
| Multi-Head(1) | $38.66 \pm 0.34$ | $31.78 \pm 0.37$ |
| Multi-Task(1) | $40.14 \pm 0.38$ | $33.62 \pm 0.38$ |
| MAML (2017)(1) | $49.31 \pm 0.40$ | $52.45 \pm 0.48$ |
| ANIL (Almost No Inner Loop)(1) | $50.23 \pm 0.42$ | $52.69 \pm 0.47$ |
| BOHI (Body Outer loop, Head Inner Loop)(1) | $50.61 \pm 0.43$ | $53.60 \pm 0.48$ |
| Multi-Head(5) | $48.99 \pm 0.33$ | $41.48 \pm 0.38$ |
| Multi-Task(5) | $50.82 \pm 0.35$ | $44.94 \pm 0.39$ |
| MAML (2017)(5) | $64.77 \pm 0.36$ | $67.66 \pm 0.42$ |
| ANIL (Almost No Inner Loop)(5) | $65.98 \pm 0.38$ | $67.44 \pm 0.43$ |
| BOHI (Body Outer loop, Head Inner Loop)(5) | $66.14 \pm 0.37$ | $68.39 \pm 0.42$ |

a query set $T_b^q$, where $N$ is the number of sampled class and $K$ is the number of instances for each class. During the inner loop, with the support set $T_b^s$, we perform fast adaptation in several gradient descent steps and obtain the task-specific parameters $\theta_{T_b}^t$ where $t$ is the number of gradient descent steps, given by:

$$\theta_{T_b}^t = \theta_{T_b}^{t-1} - \alpha \nabla_{\theta_{T_b}^{t-1}} \mathcal{L}_{T_b^s}(\theta_{T_b}^{t-1}) \tag{1}$$

where $\alpha$ is the step size for inner loop and $\mathcal{L}_{T_b^s}(\theta_{T_b}^{t-1})$ denoted as the loss on the support set $T_b^s$ after $t-1$ steps. With the query set $T_b^q$, we compute the meta loss on the task-specific parameters $\theta_{T_b}^t$ and backward to update the meta-initialized parameters $\theta$, given by

$$\theta = \theta - \beta \nabla_\theta \frac{1}{B} \sum_{b=1}^{B} \mathcal{L}_{T_b^q}(\theta_{T_b}^t) \tag{2}$$

where $\beta$ is the learning rate and $B$ is the number of sampled tasks in a batch.

## 4 IMPACT OF TASK-SPECIFIC ADAPTATION

### 4.1 THE MULTI-STEP TASK-SPECIFIC ADAPTATION IS IMPORTANT.

To explore the effectiveness of MAML, Raghu et al. (2019) have conducted sufficient experiments, indicating that the network body (the representation layers) has already absorbed the high-quality features prior. During meta-testing, instead of fine tuning on the network head (the classifier), simply building the prototypes with the support set can achieve comparable performance to MAML. Raghu et al. (2019) also shows that the task specificity of head at training can facilitate feature learning and ensure good representation learning in the network body. In our work, we show that besides the task specificity of head, the multi-step adaptation is also essential, and further study the role of network body and head during meta-training. We devise several methods using different training regimes: (1) *Multi-Task*, where all the tasks simply share one common head and the model is trained in a traditional way without inner loop adaptation; (2) *Multi-Head*, where different tasks are equipped with different heads for task specificity and the model is trained in a traditional way without inner loop adaptation; (3) *Almost No Inner Loop (ANIL)*, where the network body is fixed during the inner loop; (4) *Body Outer Loop, Head Inner Loop (BOHI)*, where the network body is updated only by the outer loop and the head is adapted only during the inner loop, making the head's meta-initialized parameters unchanged. More algorithms' details can be found in Appendix B, and implementation details can be found in Appendix C.1.

Following Raghu et al. (2019), we employ the cosine similarities between prototypes and the query datapoints to evaluate the quality of features learned. As Table 1 shows, even equipped with task-specific head, the Multi-Head training still performs worse than the standard MAML algorithm by

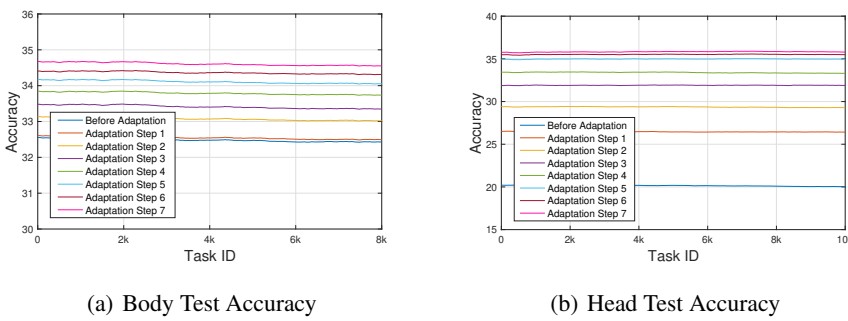

    (a) Body Test Accuracy                (b) Head Test Accuracy

Figure 1: The adaptation of the random initialized model for the sampled tasks in different steps.

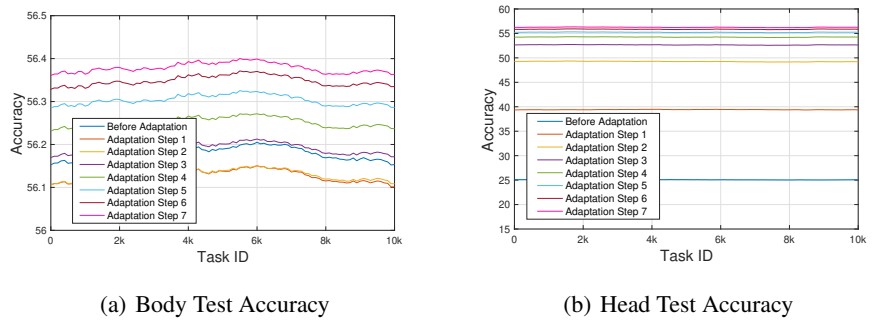

    (a) Body Test Accuracy                (b) Head Test Accuracy

Figure 2: The adaptation after 5,000 iterations for the sampled tasks in different steps .

a large margin, indicating the multi-step adaptation of MAML is helpful for features learning. The results of Multi-Head and Multi-Task show the importance of multi-step task-specific adaptation.

As the results shown in Table 1, the ANIL training remains effective comparable to the standard MAML algorithm, indicating that the task-specific adaptation of network body is unnecessary to learn good features. More interestingly, the BOHI training that keeps the meta-initialization of head unchanged even performs better than MAML, further demonstrating that good features learning depends on the multi-step task-specific adaptation of head during inner loop more than updating the meta-initialization of head in outer loop. Also, the ANIL and BOHI have similar performance, indicating that compared with learned prior knowledge in head, the inner loop adaptation, as a guidance, contributes more to the features learning. More experimental results can be found in Appendix C.2.

### 4.2 WHY IS MULTI-STEP TASK-SPECIFIC ADAPTATION IMPORTANT?

Having observed that the MAML algorithm outperforms the Multi-Task training by a large margin and the multi-step task-specific adaptation is important for features learning, we extend our analysis to explore the reason why the inner loop adaptation is essential for MAML at different stages of meta training. Specifically, we freeze the initialized MAML model and model at 5,000 iterations, sample validation tasks from the task distribution, and record the test accuracy of model in different inner loop steps. Both the body accuracy based on prototypes construction and head accuracy based on fine-tuning are given in Figure 1 and Figure 2, where "Task ID" stands for different tasks. As the results shows, at different stages of meta training, the head accuracy increases significantly in the first few adaptation steps since the model has learnt the correspondence between sample and label. However, at the beginning of training, there is only a small improvement on the body accuracy after first adaptation step. In Figure 2, as the model converges, the body accuracy even decreases in the first few adaptation steps. In the following steps, with the task-specific adaptation of head, the network body then learns better representations, further demonstrating that the multi-

---

**Algorithm 1** The Random Decision Planes (RDP) Algorithm for N-way-K-shot learning

---

**Input:** Network Body $f_\theta$, Learning Rate $\beta$, Task Distribution $P(T)$

   Perform the Gram-Schmidt method on random metrices to get the classifier set $\mathcal{P} = \{\boldsymbol{W}_i\}_{i=1}^{n_p}$

  **while** not done **do**

     Sample a batch of tasks $\{T_b\}_{b=1}^B$, where $T_b \sim P(T)$

     **for** $b \in \{1, ..., B\}$ **do**

       Sample the support set $T_b^s = \{(\boldsymbol{x}_i^s, y_i^s)\}_{i=1}^{N \times K}$

       and query set $T_b^q = \{(\boldsymbol{x}_i^q, y_i^q)\}_{i=1}^{N \times K}$ from task $T_b$.

       **for** each sample $\boldsymbol{x}$ in $\{T_b^s, T_b^q\}$ **do**

         $\boldsymbol{z} = \|f_\theta(\boldsymbol{x})\|$

       **end for**

       **define** $\mathrm{CrossEntropyLoss}(\boldsymbol{H}, \mathcal{D})$ **as** the cross entropy loss

       on the features representations set $\mathcal{D}$ with head $\boldsymbol{H}$.

       $\boldsymbol{W}^\star = \underset{\boldsymbol{W} \in \mathcal{P}}{\arg\min} \ \mathrm{CrossEntropyLoss}(\boldsymbol{W}, \{(\boldsymbol{z}_i^s, y_i^s)\}_{i=1}^{N \times K})$

       $\mathcal{L}_b = \mathrm{CrossEntropyLoss}(\boldsymbol{W}^\star, \{(\boldsymbol{z}_i^q, y_i^q)\}_{i=1}^{N \times K})$

     **end for**

     $\theta = \theta - \beta \nabla_\theta \frac{1}{B} \sum_{b=1}^B \mathcal{L}_b$

  **end while**

---

step task-specific adaptation, making the body and head have similar classification capabilities, can be regarded as a guidance to provide better gradient descent direction for the feature learning of body.

To understand this intuitive argument better, we consider a sample $(\boldsymbol{x}, y)$ for few-shot classification where the cross entropy loss is employed, formulated as:

$$\mathcal{L}_c = -\log(\frac{\exp(\boldsymbol{w}_y^\top \boldsymbol{h})}{\sum_k \exp(\boldsymbol{w}_k^\top \boldsymbol{h})}) = -\boldsymbol{w}_y^\top \boldsymbol{h} + \log(\sum_k \exp(\boldsymbol{w}_k^\top \boldsymbol{h})) \tag{3}$$

where $\{\boldsymbol{w}_1, \boldsymbol{w}_2, ..., \boldsymbol{w}_k\}$ is the weights of the classifier head, $\boldsymbol{h}$ is the body representation of $\boldsymbol{x}$. The gradients of loss $\mathcal{L}_c$ with respect to the body representation $\boldsymbol{h}$ are denoted by,

$$\frac{\partial \mathcal{L}_c}{\partial \boldsymbol{h}} = -\boldsymbol{w}_y + \frac{\sum_k \boldsymbol{w}_k \exp(\boldsymbol{w}_k^\top \boldsymbol{h})}{\sum_k \exp(\boldsymbol{w}_k^\top \boldsymbol{h})} = -\boldsymbol{w}_y + \bar{\boldsymbol{w}} \tag{4}$$

where $\bar{\boldsymbol{w}}$ is exactly the weighted average of the weights $\{\boldsymbol{w}_1, \boldsymbol{w}_2, ..., \boldsymbol{w}_k\}$. As shown in Equation 4, a reasonable direction for the network body to minimize the target loss $\mathcal{L}_c$ is to make the representation $\boldsymbol{h}$ closer to the corresponding class weight $\boldsymbol{w}_y$, given by $\boldsymbol{h} = \boldsymbol{h} + \lambda(\boldsymbol{w}_y - \bar{\boldsymbol{w}})$. As the model converges, in the first few adaptation steps, there is a significant margin between the performance of head and body, and the classifier weights contain little knowledge about correspondence between samples and labels and differences between different classes. With the low-performance head, this updating rule for body may lead to a decline in the quality of features, which also explains why the simpler BOHI, ANIL even performs better than MAML in Table 1. After several adaptation steps during the inner loop, the body then receives the useful guidance for features learning from the task-specific head since $\boldsymbol{w}_y$ can better express its corresponding class. The formulation above shows that the multi-step task-specific adaptation, making the body and head have similar classification capabilities, can provide better gradient descent direction for the features learning of body.

## 4.3 TASK-SPECIFIC ADAPTATION IN OTHER META-LEARNING ALGORITHMS

Having noticed that the multi-step task-specific adaptation of MAML, which promotes the performance of head, can facilitate the features learning of body. It works similarly for other gradient-based methods that use end-to-end fine-tuning, such as Reptile (Nichol et al. (2018)). In the case of meta-learning methods that fix the network body and only update the head during the inner loop, such as MetaOptNet (Lee et al. (2019b)) and R2-D2 (Bertinetto et al. (2018)), the convex optimization of head also aims to provide a classifier with better classification capabilities. For metric-based methods, such as Prototypical Networks (Snell et al. (2017)), the adaptation of head is actually conducted through the nearest neighbor algorithm. In conclusion, the adaptation is a common mode but

Table 2: The evaluation results of 5-way-K-shot learning for the standard MAML and Random Decision Planes (RDP) with different backbones.

| Method | Backbone | MiniImageNet-5-way | TieredImageNet-5-way | FC100-5-way |
|---|---|---|---|---|
| MAML (2017)(1) | Conv4 | $49.31 \pm 0.40$ | $52.45 \pm 0.48$ | $34.75 \pm 0.39$ |
| **RDP**(1) | Conv4 | $46.12 \pm 0.38$ | $47.63 \pm 0.44$ | $36.63 \pm 0.38$ |
| **RDP**(1) | ResNet12 | $51.16 \pm 0.43$ | $51.37 \pm 0.46$ | $37.54 \pm 0.40$ |
| MAML (2017)(5) | Conv4 | $64.77 \pm 0.36$ | $67.66 \pm 0.42$ | $43.90 \pm 0.38$ |
| **RDP**(5) | Conv4 | $63.34 \pm 0.36$ | $65.19 \pm 0.42$ | $49.46 \pm 0.39$ |
| **RDP**(5) | ResNet12 | $65.72 \pm 0.36$ | $66.31 \pm 0.41$ | $50.29 \pm 0.38$ |

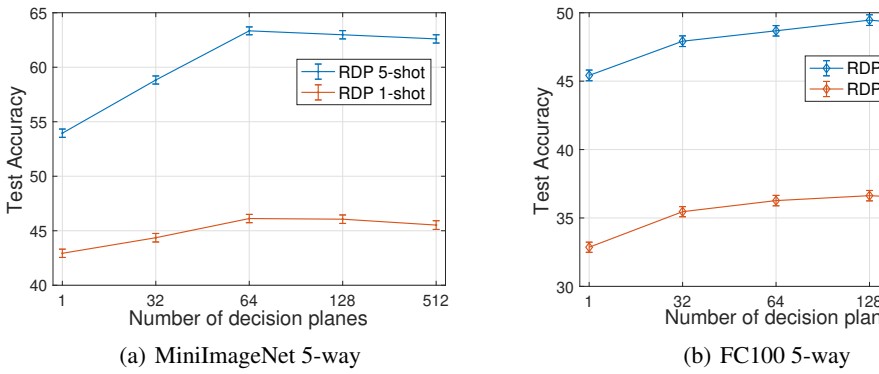

(a) MiniImageNet 5-way      (b) FC100 5-way

Figure 3: The effect of the number of decision planes on the miniImageNet and FC100 datasets.

varied in different methods. These meta-learning algorithms reveal a general formula that the inner loop is for building a task-specific head that matches the classification capabilities of body and the outer loop for task-independent features learning.

## 5 THE RANDOM DECISION PLANES ALGORITHM

As discussed above, the multi-step adaptation based on gradient descent during the inner loop aims to provide guidance for features learning of body. From this consideration, we suppose that if a suitable linear classifier is given, the feature learning can be facilitated even without gradient descent during the inner loop. From this consideration, we devise such an algorithm named Random Decision Planes (RDP), where a classifier is chosen from a predefined set $\mathcal{P}$ according to the target loss on the support set. The predefined set of classifier $\mathcal{P}$ consists of $n_p$ different orthonormal matrices that are generated through the Gram-Schmidt method from random matrices. During the inner loop, without gradient descent, we directly choose a most suitable classifier as the network head which minimizes the cross entropy loss on the support set. In the outer loop, we compute the loss based on the chosen head and run backward to update the network body. A formal description of RDP is presented in Algorithm 1. The implementation details can be found in Appendix C.1.

The overall evaluation results on three datasets are presented in Table 2. Note that we also remove the head and construct the prototypes from the body network $f_\theta$ for predictions during meta-testing. The proposed RDP algorithm performs comparably to the standard MAML method on three datasets, especially on the FC100 dataset. Without any task-specific adaptation for the network body, a best performing classifier chosen from a set of randomly generated subspaces can also be a guidance to facilitate the features learning, further suggesting that a head with better classification capabilities, is key factor to learn good representations even if the chosen approximate head performs worse than a gradient-based head, and the main purpose of task-specific adaptation is to adjust the low-performance head for features learning of body.

---

**Algorithm 2** The Meta Contrastive Learning (MCL) Algorithm for N-way learning

---

**Input:** Network Body $f_\theta$, Projection Layer $g_\phi$, Learning Rate $\beta$,
   Constant $\tau$, Task Distribution $P(T)$
   **while** not done **do**
      Sample a batch of tasks $\{T_b\}_{b=1}^B$, where $T_b \sim P(T)$
      **for** $b \in \{1, ..., B\}$ **do**
         Sample the query set $T_b^q$ from task $T_b$ where $T_b^q = \{(\boldsymbol{x}_i^q, y_i^q)\}_{i=1}^{2N}$,
         and $y_{2k-1}^q = y_{2k}^q$ where $k \in \{1, ..., N\}$.
         **for** $i \in \{1, ..., 2N\}$ **do**
            $\boldsymbol{z}_i = g_\phi(f_\theta(\boldsymbol{x}_i^q))$
         **end for**
         **for** $i \in \{1, ..., 2N\}$ and $j \in \{1, ..., 2N\}$ **do**
            $s_{i,j} = \boldsymbol{z}_i^\top \boldsymbol{z}_j / (\|\boldsymbol{z}_i\| \|\boldsymbol{z}_j\|)$
         **end for**
         **define** $l(i,j) = -\log\left(\frac{\exp(s_{i,j}/\tau)}{\sum_{k=1}^{2N} \mathbb{1}_{[k \neq i]} \exp(s_{i,k}/\tau)}\right)$
         $\mathcal{L}_b = \frac{1}{2N} \sum_{k=1}^N [l(2k-1, 2k) + l(2k, 2k-1)]$
      **end for**
      $\theta = \theta - \beta \nabla_\theta \frac{1}{B} \sum_{b=1}^B \mathcal{L}_b$
      $\phi = \phi - \beta \nabla_\phi \frac{1}{B} \sum_{b=1}^B \mathcal{L}_b$
   **end while**

---

Also, we conduct experiments to explore the impact of the number of decision planes. Results are shown in Figure 3 on two datasets. With a small set of decision planes, it can be more difficult to find a suitable head to guide the features learning, while with enough decision planes, the performance then reaches the upper limit.

## 6   THE META CONTRASTIVE LEARNING ALGORITHM

We have already seen that the multi-step task-specific adaptation to improve the classifier head can essentially facilitate the features learning of body. In total, prior gradient-based methods based on the cross-entropy loss proposes to learn the correspondence between samples and assigned labels for different tasks, thus requiring the task-specific adaptation for the classifier head during inner loop. Since the task-specific head also serves for features learning of body, we wonder if we can remove the inner loop or adaptation, and make full use of the labels information in other way to be a guidance for features learning. From this consideration and inspired by recent works (Chen et al. (2020); He et al. (2020)) about self-supervised contrastive learning, we further devise the Meta Contrastive Learning (MCL) algorithm that directly removes the inner loop and exploits the inter-sample relationship with only the query set.

Specifically, rather than using cross entropy loss for task-specific adaptation, we simply impose that normalized representations from the same class are closer together than representations from different classes. For $N$-way few-shot learning, we sample two examples per class to build the query set. Next, for a given anchor example, the meta contrastive loss pulls it closer to the point of same class while pushes the anchor farther away from the negative examples of other classes. Following Chen et al. (2020), we also employ a small neural network projection layer that maps the body features to the space where contrastive loss is applied. A formal description of MCL is presented in Algorithm 2. The implementation details can be found in Appendix C.1.

During meta-testing, we discard the projection layer $g_\phi$ and construct the prototypes from the body network $f_\theta$ for predictions. The overall evaluation results on the MiniImageNet, TieredImageNet and FC100 datasets are presented in Table 3. Note that TADAM (Oreshkin et al. (2018)) employs a extra task embedding network (TEN) block to predict element-wise scale and shift vectors, and MetaOptNet (Lee et al. (2019b)) proposes to learn a linear support vector machine (SVM) as classifier head during the inner loop. Unlike those methods, our MCL method is arguably simpler. By exploiting the relationship between different samples, we are able to remove the inner loop which contains a complex adaptation process, and devise a contrastive loss to train the network body directly. As the results shows, our method outperforms almost previous well-designed methods and

Table 3: The evaluation results of 5-way-K-shot learning for the Meta Contrastive Learning (MCL) and other baselines with different backbones.

| Method | Backbone | MiniImageNet-5-way | TieredImageNet-5-way | FC100-5-way |
|---|---|---|---|---|
| MAML (2017)(1) | Conv4 | $49.31 \pm 0.40$ | $52.45 \pm 0.48$ | $34.75 \pm 0.39$ |
| **MCL**(1) | Conv4 | $50.73 \pm 0.43$ | $53.12 \pm 0.48$ | $37.73 \pm 0.38$ |
| TADAM (2018)(1) | ResNet12 | $58.50 \pm 0.30$ | — | $40.10 \pm 0.40$ |
| MetaOptNet (2019b)(1) | ResNet12 | $62.64 \pm 0.61$ | $65.99 \pm 0.72$ | $41.10 \pm 0.60$ |
| **MCL**(1) | ResNet12 | $62.14 \pm 0.43$ | $65.98 \pm 0.50$ | $41.38 \pm 0.40$ |
| MAML (2017)(5) | Conv4 | $64.77 \pm 0.36$ | $67.66 \pm 0.42$ | $43.90 \pm 0.38$ |
| **MCL**(5) | Conv4 | $66.25 \pm 0.36$ | $69.31 \pm 0.41$ | $51.29 \pm 0.39$ |
| TADAM (2018)(5) | ResNet12 | $76.70 \pm 0.30$ | — | $56.10 \pm 0.40$ |
| MetaOptNet (2019b)(5) | ResNet12 | $78.63 \pm 0.46$ | $81.56 \pm 0.53$ | $55.50 \pm 0.60$ |
| **MCL**(5) | ResNet12 | $78.34 \pm 0.33$ | $81.09 \pm 0.37$ | $56.64 \pm 0.39$ |

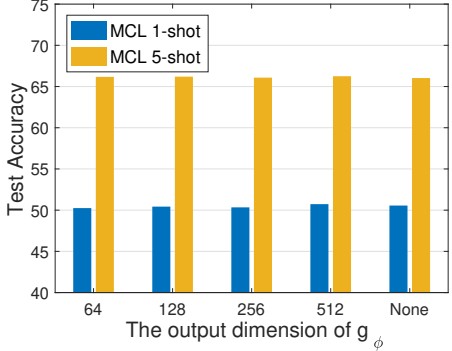

(a) MiniImageNet 5-way, Conv4

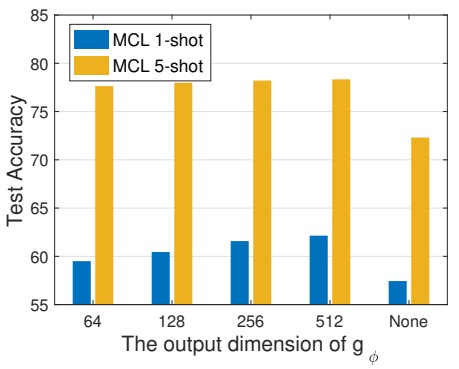

(b) MiniImageNet 5-way, ResNet12

Figure 4: The effect of output dimension of $g_\phi$ on the MiniImageNet dataset.

also achieves results comparable to MetaOptNet. More experimental results and time-efficiency analysis can be found in Appendix C.2 and C.3.

We also study the impact of the projection layer $g_\phi$. Figure 4 shows the evaluation results with different output dimensions. Note that "None" means that there is no projection layer for loss computation. As the results show, for a deeper ResNet12 backbone, the projection layer facilitates the features learning a lot ($>6\%$ for 5-shot, $>5\%$ for 1-shot). We conjecture that the projection layer is trained to extract task-specific information useful for the contrastive loss, while the body representations $h$ learns more general information. More analysis can be found in Appendix C.4.

## 7 CONCLUSION

In this paper, based on the hypothesis that feature reuse is the dominant factor for the success of MAML algorithm, we further study the impact of task-specific adaptation and devise several training regimes including BOHI, Multi-Head and so on. Also, we provide a more formal argument from the perspective of gradient descent optimization. Based on analysis above, we find that the multi-step task-specific adaptation, making the body and head have similar classification capabilities, can provide better gradient descent direction for the features learning of body. We further connect our results to other meta-learning algorithm, showing the adaptation is a common mode but varied in different methods. From our consideration, we devise the RDP algorithm where a suitable linear classifier is chosen without gradient descent and get more supporting conclusions. We also build the

MCL algorithm that removes the inner loop and exploit the inter-sample relationship, and achieve results comparable to some state-of-the-art methods.

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

## A  FEW-SHOT IMAGE CLASSIFICATION DATASETS

In this section, we introduce four benchmark datasets often used for few-shot image classification: the miniImageNet (Vinyals et al. (2016)), tieredImageNet (Ren et al. (2018)), CIFAR-FS (Bertinetto et al. (2018)) and FC100 (Oreshkin et al. (2018)).

The **miniImageNet** (Vinyals et al. (2016)) dataset is standard benchmark for few-shot image classification, comprises 100 classes randomly chosen from the original ImageNet (Russakovsky et al. (2015)) dataset, where 64 classes is used for meta-training, 16 classes for meta-validation and 20 classes for meta-testing. Each class contains 600 images of size $84 \times 84$. Since the original class splits are unavailable, we use the commonly-used split proposed in Ravi & Larochelle (2016).

The **tieredImageNet** (Ren et al. (2018)) dataset is another larger subset of ImageNet (Russakovsky et al. (2015)). This dataset contains 608 classes that are grouped into 34 high-level categories, where 20 categories (351 classes) are used for meta-training, 6 categories (97 classes) for meta-validation and 8 categories(160 classes) for meta-testing. All images are also size of $84 \times 84$.

The **CIFAR-FS** (Bertinetto et al. (2018)) dataset is a few-shot image classification benchmark, consisting of all 100 classes from CIFAR-100 (Krizhevsky et al. (2010)). These classes are randomly split into 64, 16, and 20 separately for meta-training, meta-validation and meta-testing. Each class contains 600 images of size $32 \times 32$.

The **FC100** (Oreshkin et al. (2018)) dataset is another benchmark derived from CIFAR-100 (Krizhevsky et al. (2010)). This dataset comprises 100 classes that are grouped into 20 high-level categories, where 12 categories (60 classes) are used for meta-training, 4 categories (20 classes) for meta-validation and 4 categories (20 classes) for meta-testing. Each class contains 600 images of size $32 \times 32$.

Table 4: The evaluation results of 5-way-K-shot learning on the MiniImageNet dataset.

| Method | Conv4, K=1 | Conv4, K=5 | ResNet12, K=1 | ResNet12, K=5 |
|---|---|---|---|---|
| MAML (Finn et al. (2017)) | $49.31 \pm 0.40$ | $64.77 \pm 0.36$ | $57.45 \pm 0.47$ | $72.70 \pm 0.35$ |
| Almost No Inner Loop (ANIL) | $50.23 \pm 0.42$ | $65.98 \pm 0.38$ | $59.43 \pm 0.44$ | $73.28 \pm 0.33$ |
| Body Inner, Head Outer (BOHI) | $50.61 \pm 0.43$ | $66.14 \pm 0.37$ | $59.69 \pm 0.46$ | $73.42 \pm 0.37$ |
| MetaOptNet (Lee et al. (2019b)) | — | — | $62.64 \pm 0.61$ | $78.63 \pm 0.46$ |
| Meta Contrastive Learning (MCL) | $50.73 \pm 0.43$ | $66.25 \pm 0.36$ | $62.14 \pm 0.43$ | $78.34 \pm 0.33$ |

# B  MORE DETAILS ABOUT ALGORITHMS

In this section, we provide further details about the training regimes and algorithms mentioned above. Note that we denote the meta parameters of the network as $\theta$ in previous sections. Considering the network is composed of the body (feature extractor) and head (classifier), we further rewrite $\theta$ as $\theta = [\theta_f, \theta_c]$, where $\theta_f, \theta_c$ is the parameters of body and head respectively. For a given task $T_b = \{T_b^s, T_b^q\}$, the meta-initialization updating of MAML can be expressed as follows:

$$\theta_f^t = \theta_f^{t-1} - \alpha \nabla_{\theta_f^{t-1}} \mathcal{L}_{T_b^s}(\theta_f^{t-1}, \theta_c^{t-1}), \ \theta_c^t = \theta_c^{t-1} - \alpha \nabla_{\theta_c^{t-1}} \mathcal{L}_{T_b^s}(\theta_f^{t-1}, \theta_c^{t-1})$$
$$\theta_f = \theta_f - \beta \nabla_{\theta_f} \mathcal{L}_{T_b^q}(\theta_f^t, \theta_c^t), \ \theta_c = \theta_c - \beta \nabla_{\theta_c} \mathcal{L}_{T_b^q}(\theta_f^t, \theta_c^t) \tag{5}$$

where $\alpha$ is the step size of the inner loop, $\beta$ is the learning rate. In our work, we devise several methods using different training regimes including Multi-Task, Multi-Head, Almost No Inner Loop (ANIL) and Body Outer Loop, Head Inner Loop (BOHI) to study the role of network body and head during meta-training.

The updating rules of Multi-Task can be expressed as follows

$$\theta_f = \theta_f - \beta \nabla_{\theta_f} \mathcal{L}_{T_b^q}(\theta_f, \theta_c), \ \theta_c = \theta_c - \beta \nabla_{\theta_c} \mathcal{L}_{T_b^q}(\theta_f, \theta_c) \tag{6}$$

For different tasks, the Multi-head has different heads for task specificity, given by

$$\theta_f = \theta_f - \beta \nabla_{\theta_f} \mathcal{L}_{T_b^q}(\theta_f, \theta_c^{T_b}), \ \theta_c^{T_b} = \theta_c^{T_b} - \beta \nabla_{\theta_c^{T_b}} \mathcal{L}_{T_b^q}(\theta_f, \theta_c^{T_b}) \tag{7}$$

where $\theta_c^{T_b}$ is the specific parameters for task $T_b$. The network body is fixed during the inner loop for ANIL, given by

$$\theta_c^t = \theta_c^{t-1} - \alpha \nabla_{\theta_c^{t-1}} \mathcal{L}_{T_b^s}(\theta_f^{t-1}, \theta_c^{t-1})$$
$$\theta_f = \theta_f - \beta \nabla_{\theta_f} \mathcal{L}_{T_b^q}(\theta_f, \theta_c^t), \ \theta_c = \theta_c - \beta \nabla_{\theta_c} \mathcal{L}_{T_b^q}(\theta_f, \theta_c^t) \tag{8}$$

For BOHI, the network body is updated only by the outer loop and the head is adapted only during the inner loop, making the head's meta-initialized parameters unchanged, given by

$$\theta_c^t = \theta_c^{t-1} - \alpha \nabla_{\theta_c^{t-1}} \mathcal{L}_{T_b^s}(\theta_f^{t-1}, \theta_c^{t-1})$$
$$\theta_f = \theta_f - \beta \nabla_{\theta_f} \mathcal{L}_{T_b^q}(\theta_f, \theta_c^t) \tag{9}$$

# C  MORE EXPERIMENTAL DETAILS

## C.1  IMPLEMENTATION DETAILS

For all training regimes, RDP and MCL, we use the Adam optimizer with weight decay of 5e-4 and the learning rate is set to 1e-3. For 4-layer convolution network with 64 filters, we flatten the output feature map of the network body, and obtain 1600-d features for miniImageNet and tieredImageNet, while 256-d features for CIFAR-FS and FC100. For ResNet12 network, we employ a global max pooling layer on the output feature map of the network body, and obtain 512-d features for four public datasets. During meta-training, we adopt horizontal flip, random crop and color (brightness,

Table 5: The evaluation results of 5-way-K-shot learning on the TieredImageNet dataset.

| Method | Conv4, K=1 | Conv4, K=5 | ResNet12, K=1 | ResNet12, K=5 |
|---|---|---|---|---|
| MAML (Finn et al. (2017)) | $52.45 \pm 0.48$ | $67.66 \pm 0.42$ | $63.05 \pm 0.50$ | $77.01 \pm 0.40$ |
| Almost No Inner Loop (ANIL) | $52.69 \pm 0.47$ | $67.44 \pm 0.42$ | $63.03 \pm 0.49$ | $76.98 \pm 0.41$ |
| Body Inner, Head Outer (BOHI) | $53.60 \pm 0.48$ | $68.39 \pm 0.42$ | $63.20 \pm 0.51$ | $77.11 \pm 0.40$ |
| MetaOptNet (Lee et al. (2019b)) | — | — | $65.99 \pm 0.72$ | $81.56 \pm 0.53$ |
| Meta Contrastive Learning (MCL) | $53.12 \pm 0.48$ | $69.31 \pm 0.41$ | $65.98 \pm 0.50$ | $81.09 \pm 0.37$ |

Table 6: The evaluation results of 5-way-K-shot learning on the CIFAR-FS dataset.

| Method | Conv4, K=1 | Conv4, K=5 | ResNet12, K=1 | ResNet12, K=5 |
|---|---|---|---|---|
| MAML (Finn et al. (2017)) | $61.96 \pm 0.51$ | $76.16 \pm 0.39$ | $67.41 \pm 0.51$ | $80.94 \pm 0.37$ |
| Almost No Inner Loop (ANIL) | $63.27 \pm 0.52$ | $77.25 \pm 0.38$ | $69.23 \pm 0.50$ | $81.85 \pm 0.36$ |
| Body Inner, Head Outer (BOHI) | $63.58 \pm 0.52$ | $77.11 \pm 0.38$ | $68.41 \pm 0.52$ | $80.86 \pm 0.37$ |
| MetaOptNet (Lee et al. (2019b)) | — | — | $72.00 \pm 0.70$ | $84.20 \pm 0.50$ |
| Meta Contrastive Learning (MCL) | $64.59 \pm 0.52$ | $77.24 \pm 0.38$ | $71.17 \pm 0.49$ | $84.18 \pm 0.35$ |

Table 7: The evaluation results of 5-way-K-shot learning on the FC100 dataset.

| Method | Conv4, K=1 | Conv4, K=5 | ResNet12, K=1 | ResNet12, K=5 |
|---|---|---|---|---|
| MAML (Finn et al. (2017)) | $34.75 \pm 0.39$ | $43.90 \pm 0.38$ | $38.43 \pm 0.39$ | $50.85 \pm 0.38$ |
| Almost No Inner Loop (ANIL) | $37.49 \pm 0.38$ | $49.58 \pm 0.39$ | $38.60 \pm 0.40$ | $50.70 \pm 0.39$ |
| Body Inner, Head Outer (BOHI) | $37.15 \pm 0.41$ | $48.68 \pm 0.39$ | $38.63 \pm 0.40$ | $50.65 \pm 0.38$ |
| MetaOptNet (Lee et al. (2019b)) | — | — | $41.10 \pm 0.60$ | $55.50 \pm 0.60$ |
| Meta Contrastive Learning (MCL) | $37.73 \pm 0.38$ | $51.29 \pm 0.39$ | $41.38 \pm 0.40$ | $56.64 \pm 0.39$ |

contrast, and saturation) jitter data augmentation as proposed in Gidaris & Komodakis (2018); Qiao et al. (2018). We train all models 100 epochs and take 500 batches per epoch. For MAML, BOHI and ANIL, both models are trained using 5 gradient steps of size $\alpha = 0.01$ for Conv4 and $\alpha = 0.1$ for ResNet12. For the Random Decision Planes algorithm, the number of decision planes $n_p$ is set to 64. For the Meta Contrastive Learning (MCL) algorithm, we apply a two-layer nonlinear projection layer with hidden size of 512. Also, the query datapoints come from 10 different classes for each sampled task, which is helpful for accelerating model convergence.

## C.2 More results for BOHI, ANIL, MAML, MCL

In this section, we provide complete experimental results for BOHI, ANIL, MAML and MCL with different backbones on four datasets. The complete results on four datasets are presented in Table 4, Table 5, Table 6 and Table 7 respectively. The results can further verify our description mentioned above. Good features learning depends on the multi-step task-specific adaptation of head during the inner loop more than updating the meta-initialization of head in outer loop. With the low-performance head, the update of body may even lead to a decline in the quality of features. In addition, the results on four datasets further demonstrate the effectiveness of our proposed MCL algorithm.

## C.3 The time-efficiency analysis for BOHI, ANIL, MAML, MCL

It is obvious that ANIL, BOHI and MCL can speeds up training. The results about the comparison of computation time are presented in Table 8. We implement our methods based on PyTorch and the

Table 8: The computation time of different methods.(tasks/sec)

| Method | Conv4 | ResNet12 |
|--------|-------|----------|
| MAML (Finn et al. (2017)) | 10.60 | 2.24 |
| Random Decision Planes (RDP) | 31.24 | 6.88 |
| Almost No Inner Loop (ANIL) | 30.98 | 6.86 |
| Body Inner, Head Outer (BOHI) | 31.12 | 6.88 |
| Meta Contrastive Learning (MCL) | 63.76 | 30.96 |

training of models is run on two NVIDIA 1080Ti GPU. Notice that our MCL can run much faster than BOHI and ANIL while achieves better evaluation results. The training speedups also illustrate the significant computational benefit of MCL and prove its effectiveness.

Table 9: The evaluation results about the quality of features extracted by the network body and projection layer. (5-way-5-shot on MiniImageNet, ResNet12)

| Hidden size of $g_\phi$ | Network Body $f_\theta$ | Projection Layer $g_\phi$ |
|-------------------------|-------------------------|---------------------------|
| 64 | $77.69 \pm 0.33$ | $61.83 \pm 0.37$ |
| 256 | $77.98 \pm 0.34$ | $63.46 \pm 0.38$ |
| 512 | $78.34 \pm 0.33$ | $66.27 \pm 0.40$ |

### C.4 ABOUT THE PROJECTION LAYER OF MCL

We have found that with a deeper backbone, the features learning can be facilitated a lot by the projection layer. We further evaluate the quality of features extracted by the network body and the projection layer. The evaluation results are given in Table 9. Even if the contrastive loss is applied to the projection layer, the network body learns better and general representations. We conjecture that during the meta-training, the projection layer may absorb more task-specific information while the backbone tends to learn task-independent representations.

