# OpenReview forum: "To Learn Effective Features: Understanding the Task-Specific Adaptation of MAML"
_ICLR.cc/2021/Conference — Reject_

### Official Review · AnonReviewer2 · 2020-10-27

**Rating:** 5
**Confidence:** 3

**Review:**

This paper adds to a series of papers that look at understanding how and why MAML works as well as it does, and what exactly is going on in the adaptation process. There are many open questions in this regard, and so I think the topic of the paper is interesting and of interest to the community. However I found the paper somewhat difficult to read and I'm not entirely sure what to take away from the paper. It was often not clear to me why certain things were introduced or why certain experiments were conducted. Overall I think that the contribution is not strong enough for publication at ICLR.

I'll structure my questions according to the contributions listed in the introduction, and the corresponding main sections in the paper:

Contribution 1, Section 4:
- If I understand this point correctly, then you're arguing that the task-specific adaptation in the inner loop is *necessary* such that the outer loop can learn a good feature extractor, is that correct? But doesn't the fact that you can do MLP contradict this statement, because here you've basically removed the task-specific adaptation.
- Section 4.1.: You compare four methods here, to show that multi-step task adaptation is important.
 - What do you mean with "Muti-step" adaptation? Are you focusing on the fact that you need multiple *gradient* updates in the inner loop and that one is not enough?
 - (Model 1) How is the multi-task model trained? You say that all tasks use the same head, so how do you add the task-specific information to the classifier? How does this work in the few-shot learning setting? How do you adapt the model at test time?
 - (Model 2) How is the multi-head model trained? I can see how you can do this for a small, fixed number of training tasks, where you just learn one head per task over the entire course of meta-training. But the what happens at test time? You say there is no inner loop, but the test accuracy is larger than random. So do you re-initialise a random new head and perform gradient updates on that?
 - (Model 3) Add a citation here to make clear that ANIL is from Raghul et al. 2019
 - (Model 4) Is this your own method, or is it taken from prior work? If I understand correctly a similar model is investigated in concurrent work: https://arxiv.org/abs/2008.08882 (which is not a problem; but you might be interested in what they find and it would be good to just add a brief mention of this concurrent work somewhere in the paper).
- Section 4.2: In this section you ask *why* multi-step adaptation is important.
 - In Figure 1 and 2, how come the accuracy before adaptation is significantly larger than random?
- What do you mean with "We present _sufficient_ experiments"? Sufficient in what sense?

Contribution 2, Section 5:
- It's kind of cool that the suggested algorithm of using (random) pre-defined classifiers (last layer of the network) instead of adapting the last layer using gradient descent in the inner loop works well on few-shot image classification (on two datasets worse, on one dataset better than MAML). I think this illustrates a similar point as ANIL (Raghu et al. 2019), namely that only adapting the last layer is sufficient on some problems.
-  In Figure 3 you show the number of "decision planes" in the set of classifiers that you use. I don't understand how using 1 decision plane can give such high (~30-55%) accuracy, can you explain this? If I have only one decision plane to choose from in the inner loop, then I always have to choose the same one, so adaptation isn't really possible. Shouldn't I just get random performance, so 20% on the 5-way problem then?
- I think it would help to explain in this section why you are proposing this algorithm. Is it to drive home an observation about MAML? If so what does this example do that wasn't shown in ANIL (Raghu et al. 2019)? And/or is it to propose a new method that people should use - in which case it would be good to argue why. Current meta-learning methods perform much better than what RDP can achieve and evaluating the accuracy on all available heads might be a computational overhead. But maybe there are situations where we'd rather do that than gradient updates?
- Do you have any intuition as to why RDP works better on the FC100 dataset compared to MAML, but not on the other datasets? Does that dataset have different properties? I think the reader could benefit from your insights here.
- Why does the accuracy drop slightly when 512 decision planes are used compared to using 128?

Contribution 3, Section 6:
- Do I understand correctly that this method basically does something like NIL (Raghu et al. 2019) but then explicitly trains the features with a contrastive loss such that features from different classes get pushed further apart? If that is the case, shouldn't you compare to NIL here?
- There are a few other methods that use features to compare classes, or contrastive losses, and I'm surprised you don't compare to these. E.g. https://arxiv.org/1807.02872 https://arxiv.org/2008.09942 https://arxiv.org/abs/1606.04080
- You write that "out method outperforms almost previous [sic] well-designed methods and also achieves results comparable to MetaOpNet". But current methods SOTA that aren't listed in your comparison get much higher (+20 percentage points) accuracies (https://paperswithcode.com/sota/few-shot-image-classification-on-mini-1). Could you explain why you chose to compare to the methods in Table 3?
- I think I have the same question as my last point above for [2]. It's not clear to me what the purpose of the proposed method is - illustrating a point or proposing a useful new algorithm?

---------------------------------------------------
---------------------------------------------------
UPDATE:

I have read the other reviews and the author's response.

Thank you for the thorough answers - this cleared a lot of things up and I understand much better how the multi-head, multi-task and random decision planes work. I'll increase my score to a 5 because I've gotten more insight now with the additional information, and think that the paper raises some interesting points. Overall, I still tend towards rejection - even with the updated version, I still find the contributions of the paper difficult to tease out and evaluate, and not all claims in the paper are sufficiently backed up / analysed by experiments. I would encourage the authors to try and centre the entire paper more clearly around *one single* central message in the future, and present all experiments in this light, making sure that every claim is sufficiently backed up empirically.

---

> ### Author Response · Authors · 2020-11-18
> **Author response to Review #2 (Contr1)**
>
> Contr1, Sec4:
> - Actually, we still keep task-specific adaptation for BOHI and ANIL during meta-training while perform evaluation based on the network body without task-specific adaptation(e.g. kNN) during meta-testing, indicating that the network body has already learnt a high-quality features prior. For "Multi-Task" setting, we totally remove the adaptation during meta-training and obtain poor results, indicating that the adpatation plays an important role in features learning. Doing MLP doesn't contradict our statement. (Table 1)
> - Yes, "Multi-step" means multiple gradient updates. Notice that the "Multi-Head" with task-specific adaptation for different tasks still perform poorly. Compared with other baselines, for good features learning of body, we infer that single-step adaptation isn't enough and a well-adapted classifier with multi-step updating makes more contributions. This conclusion and reasons can be drawn according to Figure 1-2, and serves for the following section.
> - (Model 1) A formal description about "Multi-Task" can found in appendix.B. We remove the inner loop and directly use a multi-task loss by summing up the target loss of all sampled tasks. The training procedure is the same as MAML. Under this setting, no task-specific information can be added to the classifier. We perform evaluation only based on the network body(e.g. kNN) during meta-testing. As results shown in Table 1, this model doesn't work well in few-shot learning setting.
> - (Model 2) A formal description about "Multi-Head" can found in appendix.B. The training parameters(e.g. learning rate, optimizer) are the same as MAML. During meta-testing, without task-sepcific head, we perform evaluation only based on the network body(e.g. kNN). The test accuracy is larger than random since the body has learnt a features prior even if the prior is bad.
> - (Model 3) Thanks for your suggestion. Notice that the ANIL we implemented outperforms ANIL from Raghul et al. 2019.
> | Method,    | miniImageNet(1)     | miniImageNet(5) |   tieredImageNet(1)     | tieredImageNet(5) |
> | :---------:| :--------------: | :--------------: | :--------------: | :--------------: |
> | ANIL(Raghul et al. 2019) |  46.7 ± 0.4    | 61.5 ± 0.5     |  -    | -     |
> | our ANIL |  50.23 ± 0.42    | 65.98 ± 0.38     |  52.69 ± 0.47    | 67.44 ± 0.43     |
>
> - (Model 4) Thanks for your suggestion. BOIL and BOHI have a totally different behaviour. The BOIL from https://arxiv.org/abs/2008.08882 freezes the head of the model during inner loop, while our proposed BOHI updates only the head during inner loop. The conclusions reached are also different.
> - About Figure 1 and 2. The evaluation of body accuaracy is conducted based on the features extracted by the body(e.g. kNN). A random-initialized body has representation ability to some extent. For example, an identity layer just represents input itself. By contrast, for head accuaracy, the classifier is required to learn the mapping from features to labels, which can't done by initialization.

---

> ### Author Response · Authors · 2020-11-18
> **Author response to Review #2 (Contr2) (continued)**
>
> Contr2, Sec5:
> - The RDP is a natural extension from the given formulation that making the body and head have similar classification capabilities can provide better gradient descent direction for the features learning of body. This also illustrates a stronger point that the real purpose of gradient descent during inner loop is to adjust the low-performance head rather than learn good meta-initialization for fast adaptation. We can find another way to provide guidance for features learning instead of applying time-consuming gradient descent.
> - Actually, using one decision plane is similar to "Multi-Task" except that the pre-defined classifier is a more reasonable plane that is a orthonormal matrix. The accuracy is higher than "Multi-Task" as expected. Just because this setting isn't the best choice for features learning, doesn't mean the network body can't learn something from the target loss and datapoints. The given formulation also presents an explanation that the network body learns representation to minimize to the target loss. (from head)
> - The results of ANIL(Raghu et al. 2019) simply indicate the necessity of task-specific adaptation for features learning, while we further explore how the task-specific adaptation, aiming to adjust the low-performance head, provide guidance for features learning, and notice that making the body and head have similar classification capabilities is the key point. A low-performance head may even lead to a decline in the quality of features. The RDP algorithm is actually a natural extension from previous section and also a strong evidence. A more matching classifier, even task-independent and generated randomly, can facilitate the features learning. Since there is no time-consuming gradient descent and the evaluation can be parallelized, the RDP still maintains high time efficiency(Table 8).
> - As shown in Table 7, the three algorithms RDP, BOHI and ANIL work better on FC100 compared to MAML with a Conv4 architecture. The difference is that MAML updates both the body and head during inner loop. As we said, a low-performance head may even lead to a decline in the quality of features. The task-specific adaptaion for body during inner loop can be unnecessary. Since the FC100 has the smallest sample size, the features learned by MAML could be more sensitive about the impact of the head.
> - For RDP, we choose the classifier that minimizes the target loss but the chosen class weight may not be the best to describe this class of samples. More classifiers can provide more choices but may introduce disturbance for specific class(more class weights to choose). As results shown, this kind of disturbance actually has litte effect.

---

> ### Author Response · Authors · 2020-11-18
> **Author response to Review #2 (Contr3) (continued)**
>
> Contr3, Sec6:
> - Actually, the NIL(Raghu et al. 2019) totally follows the training regime of MAML during meta-training while removes the head during meta-testing. Thanks for your advice. Limited by page size, we don't compare MCL with NIL but note that the MCL outperforms all the algorithms with NIL(Raghu et al. 2019). Some results are given in the following table.
> - Thanks for your advice. Limited by page size, we don't compare to these algorithms since the MCL outperforms them by a large margin. We will add the results in the following revision. Some results are given in the following table.
> | Method,    | backbone  | miniImageNet(1)     | miniImageNet(5) |   tieredImageNet(1)     | tieredImageNet(5) |
> | :---------:| :--------------: | :--------------: | :--------------: | :--------------: | :--------------: |
> | Matching Network(https://arxiv.org/abs/1606.04080) | Conv4 |   43.56 ± 0.84    | 55.31 ± 0.73     |  -    | -     |
> | MAML-NIL(Raghu et al. 2019) | Conv4 |   48.4 ± 0.3    | 61.5 ± 0.8     |  -    | -     |
> | ANIL-NIL(Raghu et al. 2019) | Conv4 |   48.0 ± 0.7    | 62.2 ± 0.5     |  -    | -     |
> | our MCL | Conv4 |  50.73 ± 0.43    | 66.25 ± 0.36     |  53.12 ± 0.48    | 69.31 ± 0.41     |
> | our MCL | Res12 |  62.14 ± 0.43    | 78.34 ± 0.33     |  65.98 ± 0.50    | 81.09 ± 0.37     |
> - Most current SOTA methods are acutually transfer-based learning rather than meta learning(e.g. SKD), or absorb extra training data(e.g. AmdimNet). The well-designed MCT meta-learning approach assume we can access other unlabeled query examples, which is called transductive learning. Considering the purity of these methods, we finally choose the MetaOptNet as the baseline. We present more results compared with those meta-learning approaches.
> | Method,    | backbone  | miniImageNet(1)     | miniImageNet(5) |   tieredImageNet(1)     | tieredImageNet(5) |
> | :---------:| :--------------: | :--------------: | :--------------: | :--------------: | :--------------: |
> | SimpShot[2] | Conv4 |  49.69    | 66.92     |  __   |  __     |
> | our MCL | Conv4 |  50.73 ± 0.43    | 66.25 ± 0.36     |  53.12 ± 0.48    | 69.31 ± 0.41     |
> | SimpShot[2] | Res18 |  62.85    | 80.02     | 69.09 ± 0.22   |  84.58 ± 0.16     |
> | our MCL | Res12 |  62.14 ± 0.43    | 78.34 ± 0.33     |  65.98 ± 0.50    | 81.09 ± 0.37     |
> | R2-D2+Task Aug[3] | Res12 |  62.32 ± 0.45    | 78.81 ± 0.34     |  -    | -   |
> | FEAT, pretrained backbone | Res18 |  66.78     | 82.05     |  70.80 ± 0.23    | 84.79 ± 0.16     |
> | our MCL, pretrained backbone | Res12 |  67.54 ± 0.41    | 83.23 ± 0.32     |  71.42 ± 0.52    | 85.89 ± 0.35     |
> | Transductive BD-CSPN[4] | WRN-28-10 |  70.31 ± 0.93    | 81.89 ± 0.60     |  78.74 ± 0.95    | 86.92 ± 0.63     |
>
> [1] Ye et al., Few-Shot Learning via Embedding Adaptation with Set-to-Set Functions. CVPR 2020.
>
> [2] Wang, Yan, et al. "Simpleshot: Revisiting nearest-neighbor classification for few-shot learning." arXiv preprint arXiv:1911.04623 (2019).
>
> [3] Liu, Jialin, Fei Chao, and Chih-Min Lin. "Task Augmentation by Rotating for Meta-Learning." arXiv preprint arXiv:2003.00804 (2020).
>
> [4] Liu, Jinlu, Liang Song, and Yongqiang Qin. "Prototype Rectification for Few-Shot Learning." arXiv preprint arXiv:1911.10713 (2019).
>
> - The task-specific adaptation aims to promote the performance of head and finally facilitate features learning. It works similarly for other gradient- based methods that use end-to-end fine-tuning, such as Reptile (Nichol et al. (2018)). In the case of meta-learning methods that fix the network body and only update the head during the inner loop, such as MetaOptNet (Lee et al. (2019b)) and R2-D2 (Bertinetto et al. (2018)), the convex optimization of head also aims to provide a classifier with better classification capabilities. For metric-based methods, such as Prototypical Networks (Snell et al. (2017)), the adaptation of head is actually a nearest-neighbor head. The adaptation may be time-consuming (gradient descent). As an natural extension, we attempt to make full use of the labels information and relationship between samples to directly provide guidance for network body. We don't have to consider if the head adapts well for features learnig, and can pay more attention to the network design. In total, the MCL is an extension of prior points and also a useful new algorithm for meta learning.

---

### Official Review · AnonReviewer4 · 2020-10-28
**This paper analyzes the impacts of task-specific head and model body in meta-learning. New algorithms are proposed based on the analysis. Some improvements are obtained using the proposed methods.**

**Rating:** 4
**Confidence:** 3

**Review:**

Strengths:

1.	Based on previous work (Raghu et al. 2019), this paper provides some analysis to explain why multi-step task-specific adaptation is important.

2.	The idea of introducing contrastive learning is reasonable and leads to performance improvement in experiments.

Weaknesses:

1.	The proposed methods have little improvements over base models; Random Decision Planes (RDP) actually obtains worse results than the basic MAML. Although it is reasonable for the proposed method to use contrastive learning, its novelty is quite limited. Some questions are listed below, which would be great if they can be answered.

2.	Analyses in some section are of heuristic nature, and thus not very convincing (please see the detailed comments below).

3.	The presentation of the paper needs some improvement. Since the proposed algorithms are mainly based on the idea of learning better feature extraction, the analysis on why multi-step adaptation is important, thus, contributes little to the understanding of the proposed methods, which can be directly motivated by results in Raghu et al. 2019.

Detailed Comments & Questions:

1.	Would it be possible to provide some more details on the implementation of ‘Multi-Head’ in section 4.1? How to make the head parameters task-specific to randomly sampled tasks? Are the heads re-initialized at every iteration? Or simply use n heads for a batch of n tasks, and then apply the updated n heads to the next batch of n tasks? I think this might be important.

2.	I do agree that low-accuracy head may provide bad signals for the model body. But the heuristic nature of the analysis is not convincing enough. Firstly, the body is updated based on the head’s prediction accuracy.  Obviously, there will be a significant gap between the body accuracy and the head accuracy. Secondly, the parameters in the body might not change “efficiently” as the head parameters, because the gradient with respect to the body parameters (especially the low-level features) might be small. It will be helpful if we can know the averaged norm of elements changes (or gradients) in the body and the head, respectively, or the averaged spectral norm of the matrix changes (or gradients) of the body and the head, respectively. Also, what if different step sizes are used in the body and the head? Overall, question about the importance of multi-step adaptation is still not well-answered.

3.	In Table 2, results of RDP are shown, which are worse than  the basic MAML. If the analysis in previous sections are correct, why do not simply use ANIL or BOHI, which totally avoid the influence of the low-accuracy heads on the model body and  the results after adaptation, and obtain much better experimental results? What is the benefit of using RDP?

4.	Is it possible to provide more details on meta-testing for MCL? Is the classification based on RDP or using a new fully connected layer (as head)? Will the body also be updated during meta-testing? What will the results be if using cosine similarity as in Raghu et al. 2019, which might be better to illustrate the quality of features.

5.	Is it possible to visualize the features using algorithms like t-SNE? This will allow us to see how the contrastive learning leads to features that can better distinguish/discriminate samples from different classes.

6.	I agree that contrastive learning could benefit feature extraction, and consequently helps the final results after adaptation. It might be interesting to see whether multiple datasets can be used for pre-training, e.g. train the model by contrastive loss using FC100, CIFARFS, and ImageNet 32\times 32, and then test on only one of them.

---

> ### Author Response · Authors · 2020-11-18
> **Author response to Review #4**
>
> 1. Thanks for your suggestion. For "Multi-Head", we employ n heads for a batch of n tasks and re-initialize these heads for next batch of tasks to keep head parameters completely task-specific to randomly sampled tasks.
> 2. The results shown in Figure 1-2 can be also a strong evidence. At the beginning of training, there is only a small improvement on the body accuracy after first adaptation step(a gap between head and body). As the model converges, the body accuracy even decreases in the first few adaptation steps since a bigger gap may lead to a decline in the quality of features(given by the formulation). To know how the body change as the head paramters, we further provide the experiment results about the cosine similarity between the body representation and its corresponding class weight after adaptation. Note that we set the step size as 0.1 to make the results more obvious. The body representations are moved closer to the class weights. Since the classifier weights contain little knowledge about correspondence between samples and labels and differences between different classes, the first few steps may lead to a decline.
> | Method, | Before Adap, | After Step 1, | After Step 2 |
> | :-------:  | :---------------------------------:| :--------------------------------: | :-----------------------------: |
> | MAML, it0   |  -0.01 | 0.08 | 0.14 |
> | MAML, it1000|  -0.01 | 0.24 | 0.29 |
>
> Moreover, notice that the BOHI can be seen as model with 0 step for body and 5 steps for head. To make our argument more convincing, we coduct more experiments with different step sizes for body and head. As the results shown, the key point for better features learning is the adaptation of head. Due to the gap between body and head, the final performance even drops slightly with step size > 0 for network body.
>
> | body    | head | miniImageNet(1)     | miniImageNet(5) |
> | :---------:| :---------: | :--------------: | :--------------: |
> | 0 | 5 |  50.61 ± 0.43    | 66.14 ± 0.37     |
> | 1 | 5 |  50.32 ± 0.45    | 65.88 ± 0.36     |
> | 3 | 5 |  49.86 ± 0.43    | 65.21 ± 0.38     |
> | 5 | 5 |  49.31 ± 0.40    | 64.77 ± 0.36     |
>
> 3. Actually, the RDP is a natural extension from the given formulation that making the body and head have similar classification capabilities can provide better gradient descent direction for the features learning of body. This also illustrates a stronger point that a more matching classifier, even task-independent and generated randomly, can facilitate the features learning. We can find another way to provide guidance for features learning instead of applying time-consuming gradient descent.
> 4. For MCL, during meta-testing, we also remove the head, freeze the network body and use the learned features for effective evaluation(building the prototypes with the support set). As you said, we think it might be better to illustrate the quality of features.
> 5. Thanks for your advice. We visual the feature embeddings using t-SNE for MAML and MCL. The results are presented as a supplementary file named as "t-SNE.pdf".  As the figure shown, the features extracted by MCL have better clustering compared to MAML, which also indicates the effectiveness of MCL.
> 6. Thanks for your advice. We conduct more experiments on cross-domain adaptation(pretraining with another dataset) to further verify the effectiveness of our methods. For MCL, during meta-training, we fine tune the network with the support set. The evaluation results are presented as follow:
> | Method, | tieredImageNet -> miniImageNet, | miniImageNet -> tieredImageNet, |
> | :-------:  | :----------------------------------:| :----------------------------------: |
> | MAML(1)   | 49.83 ± 0.37   |  52.12 ± 0.36   |
> | ANIL(1)   | 50.23 ± 0.36   |  52.24 ± 0.38   |
> | MCL(1)    | 51.73 ± 0.43   |  53.36 ± 0.45   |
> | MAML(5)   | 64.72 ± 0.36   |  64.35 ± 0.35   |
> | ANIL(5)   | 64.99 ± 0.38   |  64.44 ± 0.37   |
> | MCL(5)    | 69.25 ± 0.42   |  68.22 ± 0.41   |

---

### Official Review · AnonReviewer3 · 2020-10-28

**Rating:** 5
**Confidence:** 4

**Review:**

**Summary:**
* This paper conducts analysis of the task-specific adaptation in the MAML algorithm, building on recent work analysing the effectiveness of MAML. The paper explores different variants of MAML, and provides empirical analysis to argue that the multi-step task-specific adaptation of a network head in the MAML algorithm is important in learning good representations and thus enabling effective few-shot learning performance.
* The paper then proposes two algorithms: one that removes gradient-based updates for a network head (instead using random matrices for classification) and another one that uses contrastive learning (no inner loop adaptation). These are evaluated on benchmark datasets, and shown to have improved computational speed.


**Overall Comments:** This paper consists of some analysis on task-specific adaptation, and presentation of two related few-shot learning methods that build on some of the insights from the analysis. As specified in the detailed comments, I am unconvinced about some of the experimental evaluation in the analysis section, and I am not sure the comparisons are fair.
Relatedly, one of the two algorithms presented is also unconvincing to me (with the presented evaluation, it doesn't appear to strengthen the claim made by the analysis).
In contrast, the second algorithm presented is clear and interesting, with notable computational benefits and performance improvements over baselines.
A stronger version of this paper could either focus on the analysis, making the claims clearer and strengthening the evaluation (see thoughts below), or present a shortened version of the analysis and then focus more on the MCL algorithm, stressing its performance improvements and computational benefits. In its current form, due to a weaker analysis section and limited analysis of MCL, I do not think this paper is strong enough for acceptance.


**Detailed Comments:**
* The analysis about the importance of multi-step adaptation is a bit unclear at points, and the experimental evaluation may not support the claim that the multi-step task specific adaptation is important. If the claim is the multi-step adaptation in MAML is specifically the important aspect, then a fairer comparison would be MAML/ANIL/BOHI with 1 inner adaptation step for the head vs MAML/ANIL/BOHI with 5 inner adaptation steps for the head, and comparing how these affect feature learning.
* As explored in Raghu et al (2019), there is an alignment problem that could make the multi task setting perform poorly, so this result is not as surprising.
* One aspect of the multi head setting is a bit unclear -- for every task, do we initialise a new 5-way head and then perform only 1 gradient step to update these parameters? It seems that a fairer comparison would be to perform more than 1 update step. From a randomly initialised head, I would expect > 1 step to be important to help classification performance. Again though, if the claim is specifically about multi-step adaptation being important for feature learning in MAML, then the 1 inner step vs 5 inner steps on MAML/ANIL/BOHI is a more relevant comparison, as the multi head situation is fundamentally different.
* Section 4.2 is a more convincing argument (both the mathematical statement and the plots) of the important of task specific adaptation of the head. I think this analysis as a whole could be summarised as how good gradient signal for the network body is only obtained when the head of the network is (at least somewhat) effective at classification, and a bad head will lead to poorer learning signal. I also want to note that in Fig 2, the improvements in the body's representations is quite small overall (a delta of < 0.3 percent). Minor comment: it would be good to specify in the text/figure caption some more experimental details about Figs 1/2 (dataset, N way K shot, etc).
* Section 4.3 appears to extend the statement made in Section 6 of Raghu et al (2019) to newer methods, but is still an interesting observation. Minor comment: The method in Lee et al (2019b) is MetaOptNet, not MetaOpNet.
* The RDP algorithm is an interesting investigation, but I am not convinced of its utility or that it supports the overall argument of the paper. Even though FC100 performance is improved, the performance on Mini/TieredImageNet is appreciably worse than MAML, especially for 1 shot settings. Further evaluation would help this claim.
* The MCL algorithm section was interesting and I was impressed by the results, and the computational speedups that this method allows for. I think that this method itself is a good contribution. Table 8 in the appendix is in my opinion interesting enough that it would be good to put it in the main text. It is a nice idea to apply the insights from contrastive learning for good representations to FSL, especially as prior work has demonstrated good representation learning to be very important for good FSL performance.

---

> ### Author Response · Authors · 2020-11-18
> **Author response to Review #3**
>
> - Actually, we can regard MAML/ANIL/BOHI as a special Multi-Head model where each head is initialized with the same paramters and more than one gradient step is taken. Also, the "Multi-Head" model is equipped with task-specific parameter compared to "Multi-Task" model. Both the Multi-Head and Multi-Task obatin poorer results than MAML/ANIL/BOHI. The evaluation results of BOHI, Multi-Head and Multi-Task, can indicate that only equipped with the task-specific parameters is not enough to learn better features, and the multi-step adaptation is the key point. And Thanks for your advice, to make our argument more convincing, we provide extra evaluation results.
> | Method,    | adaptation step  | miniImageNet(1)     | miniImageNet(5) |   tieredImageNet(1)     | tieredImageNet(5) |
> | :---------:| :--------------: | :--------------: | :--------------: | :--------------: | :--------------: |
> | ANIL | 1 |  41.89 ± 0.43    | 54.06 ± 0.37     |  35.74 ± 0.48    | 49.44 ± 0.42     |
> | ANIL | 3 |  46.88 ± 0.41    | 61.96 ± 0.38     |  49.82 ± 0.46    | 63.25 ± 0.42     |
> | ANIL | 5 |  50.23 ± 0.42    | 65.98 ± 0.38     |  52.69 ± 0.47    | 67.44 ± 0.43     |
> | BOHI | 1 |  42.37 ± 0.41    | 54.56 ± 0.39     |  36.12 ± 0.46    | 50.63 ± 0.43     |
> | BOHI | 3 |  47.28 ± 0.43    | 62.98 ± 0.38     |  50.51 ± 0.45    | 63.82 ± 0.41     |
> | BOHI | 5 |  50.61 ± 0.43    | 66.14 ± 0.37     |  53.60 ± 0.48    | 68.39 ± 0.42     |
>
> - Yes, "Multi-Task" does suffer from the alignment problem that leads to poor performance. Notice that the implement "Multi-Task" still outperforms the one in Raghu et al (2019) by a large margin. The results indicate that even without the task specificity at training, feature learning can still be done. However, to achieve better results, adaptation that makes the body and head have similar classification capabilities is the key point.
> | Method,    | miniImageNet(1)     | miniImageNet(5) |
> | :---------:| :--------------: | :--------------: |
> | Multi-Task[1] |  26.50 ± 1.10    | 34.20 ± 3.50     |
> | our Multi-Task|  40.14 ± 0.38    | 50.82 ± 0.35     |
>
> - As you said, we initialize a new 5-way head and then perform only one gradient step to update these parameters(both body and head). Also, we can regard MAML/ANIL/BOHI as a special multi-head model where each head is initialized with the same paramters and more than one gradient step is taken.
> - Thanks for your advice. Yes, as you said, the improvements in the body's representations is quite small overall. Empirically, as the model converges, the performance rises more slowly.
> - Thanks for the reminder. In Section4.3, we further discuss a general formula for both gradient-based and metric-based methods. Those methods(e.g. MAML, Reptile, MetaOptNet, Prototypical Networks) can be regarded as classifier-based meta learning. The support set is used for finding a suitable classifier, such as MLP for Reptile and MAML, SVM for MetaOptNet and the nearest-neighbor head for Prototypical Networks. The adaptation on the support set aims to construct a supporting decision space to facilitate the feature learning. The adaptation for those methods is also to make the body and head have similar classification capabilities.
> - Actually, the RDP is to drive an observation about meta learning. Specifically, the RDP is a natural extension from the given formulation that making the body and head have similar classification capabilities can provide better gradient descent direction for the features learning of body. This also illustrates a stronger point that a more matching classifier, even task-independent and generated randomly, can facilitate the features learning. Since the chosen approximate head performs worse than a gradient-based head, the upper bound may be lower. Moreover, we re-implement the MAML algorithm and achieve much higher performance. The prior implementation of MAML could be even worse than RDP. The results are presented as follow:
> | Method,    | backbone  | miniImageNet     | tieredImageNet   |
> | :---------:| :--------:| :--------------: | :--------------: |
> | MAML(1)[1] |  Conv4    | 48.47 ± 0.26     |  48.80 ± 0.34    |
> | RDP(1)     |  Conv4    | 46.12 ± 0.38     |  47.63 ± 0.44    |
> | our MAML(1)|  Conv4    | 49.31 ± 0.40     |  52.45 ± 0.48    |
>
> | Method,    | backbone  | miniImageNet     | tieredImageNet   |
> | :---------:| :--------:| :--------------: | :--------------: |
> | MAML(5)[1] |  Conv4    | 60.36 ± 0.25     |  64.27 ± 0.27    |
> | RDP(5)     |  Conv4    | 63.34 ± 0.37     |  65.19 ± 0.42    |
> | our MAML(5)|  Conv4    | 64.77 ± 0.36     |  67.66 ± 0.42    |
>
> | Method,    | backbone  | miniImageNet     |
> | :---------:| :--------:| :--------------: |
> | MAML(5)[1] |  Res12    | 67.96 ± 0.28     |
> | RDP(5)     |  Res12    | 65.72 ± 0.36     |
>
> - Thanks for your suggestion. We will put it in the main text for the following revision.
>
> [1] Does MAML really want feature reuse only? Jaehoon Oh, Hyungjun Yoo, ChangHwan Kim, Se-Young Yun

---

### Official Review · AnonReviewer1 · 2020-10-28
**Official Blind Review #1**

**Rating:** 3
**Confidence:** 5

**Review:**

Summary:

In this paper, the authors investigate the inner-loop optimization mechanism of meta-learning algorithms. The analysis shows the effectiveness of the multi-step adaptation and (1) the key of meta-learning is how to design a well-differentiated classifier. They then propose Random Decision Planes (RDP) and Meta Contrastive Learning (MCL) and achieve comparable performance with existing methods.


Pros:
1. Empirically investigating the performance w.r.t. the change of optimization mechanism on both head and body is very important in the meta-learning field.

Cons:
1. My major concern is about the contribution of this paper.
 - The discussion is interesting. But most of the analysis results is widely accepted. For example, adapting the head layer improves performance. One claim  "as the model converges, the body accuracy even decreases in the first few adaptation steps" is not well-explained.
 - The final goal of this paper is to design a better metric for meta-learning (feature mapping function + differentiated metric design). I think the goal is the same as metric-based meta-learning, which is not new for me.
 - In addition, the baselines and experiments are not sufficient to support the goal. Especially, the proposed methods (both MCL and RDP) only show comparable performance compared with metric-based methods (e.g., MetaOptNet). It would also be better if the authors can add more metric-based baselines for comparison (e.g., [1],[2]). Moreover, they should also involve metric-based methods in the efficiency comparison section (Appendix C.3).
Thus, I feel the overall contributions are not enough to be accepted.

2. Besides the Cons 1, if the authors focus on gradient-based meta-learning. Similar to ANIL, more experiments on different types of applications are supposed to conduct. For example, the experiments on regression and reinforcement learning tasks.

3. The paper is not well-written. Here are some comments:
 - It would be more clear to formally formulate the MCL and RDL with more notations and equations.
 - Most figures are not clear. It is better to replot them with larger font size in legend, thicker lines, etc.

[1] Ye et al., Few-Shot Learning via Embedding Adaptation with Set-to-Set Functions. CVPR 2020.

[2] Wang, Yan, et al. "Simpleshot: Revisiting nearest-neighbor classification for few-shot learning." arXiv preprint arXiv:1911.04623 (2019).

---

> ### Author Response · Authors · 2020-11-18
> **Author response to Review #1**
>
> 1.
> - We also concern about how task-specific adaptation improves the performance, and notice that making the body and head have similar classification capabilities can facilitate the features learning. Following the discussion, we then devise two algorithms. As for the claim "as the model converges, the body accuracy even decreases in the first few adaptation steps", we can take a look at Figure 2. Note that there is a huge gap between body and head. For the first few adaptation steps, the head is improved while the learned representations of body are worse in discriminating samples from different classes. A more formal description is also given from the perspective of gradient descent in Section4.2. This also serves for the previous argument about how task-specific adaptation improves the performance.
> - Actually, in Section4.3, we discuss a general formula for both gradient-based and metric-based methods. Those methods(e.g. MAML, Reptile, MetaOptNet, Prototypical Networks) can be regarded as classifier-based meta learning. The support set is used for finding a suitable classifier, such as MLP for Reptile and MAML, SVM for MetaOptNet and the nearest-neighbor head for Prototypical Networks and Simpleshot. The adaptation on the support set aims to construct a supporting decision space to facilitate the feature learning, and to make the body and head have similar classification capabilities. From the previous consideration, the way of the adaptation and the number steps have a deep impact on the performance. The proposed MCL isn't based on the embeddings space constructed using the support set. Instead, we directly use the inter-samples relationship to provide guidance for features learning, focus on the design of backbone network. The metric space is self-constructed using the query datapoints. With less computation costs compared to methods complex adaptation, the MCL can still achieve good results.
> - Thanks for your advice. We will add those baselines in the following revision. The proposed [2] uses a pretrained backbone through  classify all seen classes with the cross-entropy loss. To give a fair comparison, we also provide the results for MCL with pretrained backbone.
> | Method,    | backbone  | miniImageNet(1)     | miniImageNet(5) |   tieredImageNet(1)     | tieredImageNet(5) |
> | :---------:| :--------------: | :--------------: | :--------------: | :--------------: | :--------------: |
> | SimpShot[2] | Conv4 |  49.69    | 66.92     |  __   |  __     |
> | MCL | Conv4 |  50.73 ± 0.43    | 66.25 ± 0.36     |  53.12 ± 0.48    | 69.31 ± 0.41     |
> | SimpShot[2] | Res18 |  62.85    | 80.02     | 69.09 ± 0.22   |  84.58 ± 0.16     |
> | MCL | Res12 |  62.14 ± 0.43    | 78.34 ± 0.33     |  65.98 ± 0.50    | 81.09 ± 0.37     |
> | FEAT, pretrained backbone | Res18 |  66.78     | 82.05     |  70.80 ± 0.23    | 84.79 ± 0.16     |
> | MCL, pretrained backbone | Res12 |  67.54 ± 0.41    | 83.23 ± 0.32     |  71.42 ± 0.52    | 85.89 ± 0.35     |
>
> 2. Thanks for your advice. To provide more convincing conclusions, we provide more expriment results on reinforcement learning tasks. The results are presented as follows:
> | Method,    | 2D-Navigation| HalfCheetah-Velocity |
> | :---------:| :--------------: | :--------------: |
> | MAML | -20.3   | -139.0     |
> | ANIL |  -20.1    | -120.9     |
> | BOHI |  -20.8   | -141.0     |
> 3. Thanks for your suggestion. We have modified the figures to make them clearer in the new submitted revision.
>
> [1] Ye et al., Few-Shot Learning via Embedding Adaptation with Set-to-Set Functions. CVPR 2020.
> [2] Wang, Yan, et al. "Simpleshot: Revisiting nearest-neighbor classification for few-shot learning." arXiv preprint arXiv:1911.04623 (2019).

---

### Decision · Program_Chairs · 2021-01-07
**Final Decision**

**Decision:**

Reject

**Comment:**

MAML is a well-known gradient-based bi-level optimization to learn a good initialization over a set of relevant tasks. This paper investigate different variants of MAML, providing empirical analysis of two new algorithms (RDP and MCL). Reviewers agree that it is interesting to see what the change of optimization mechanism on both head and body brings to us in the MAML framework. This is done by only empirical analysis. However, all reviewers have concerns that the current version (or even revised one after the author responses) does not contain substantial contributions over existing work in the sense that: (1) experiments do not support well what's been claimed; (2) writing should be much improved to clearly explain the formulation of RDP and MCL, as well as figures in experiments; (3) the analysis about the importance of multi-step adaptation is not clear (Section 4); (4) the proposed method has little improvements over baseline methods.  Without any positive feedback from reviewers, I do not have choice but to suggest rejection.